# Role of Zirconia in Oxide-Zeolite Composite for Thiolation of Methanol with Hydrogen Sulfide to Methanethiol

**DOI:** 10.3390/nano12111803

**Published:** 2022-05-25

**Authors:** Tinglong Yang, Mengqin Yao, Jun Ma, Peng Chen, Tianxiang Zhao, Chunliang Yang, Fei Liu, Jianxin Cao

**Affiliations:** 1Department of Chemical Engineering, School of Chemistry and Chemical Engineering, Guizhou University, Guiyang 550025, China; tlyangbuaichifan@163.com (T.Y.); mqyao@gzu.edu.cn (M.Y.); jma3@gzu.edu.cn (J.M.); chenpengdeyoujian@163.com (P.C.); guicv2287@foxmail.com (T.Z.); clyang@gzu.edu (C.Y.); 2Guizhou Key Laboratory for Green Chemical and Clean Energy Technology, Guiyang 550025, China

**Keywords:** ZrO_2_ crystalline phase, NaZSM-5 modification, hydrothermal coating strategy, Si/Al ratio, methanol thiolation

## Abstract

In this paper, the molecular sieve NaZSM-5 was modified with zirconium dioxide (ZrO_2_) by a hydrothermal coating process and other methods. By comparing the effects of the crystal phase structure of ZrO_2_ and the compositing method on the physicochemical properties and catalytic performance of the obtained composites, the structure–performance relationship of these composite catalysts was revealed. The results indicate that in the hydrothermal system used for the preparation of NaZSM-5, Zr^4+^ is more likely to dissolve from m-ZrO_2_ than from t-ZrO_2_, which can subsequently enter the molecular sieve, causing a greater degree of desiliconization of the framework. The larger specific surface area (360 m^2^/g) and pore volume (0.52 cm^3^/g) of the m-ZrO_2_/NaZSM-5 composite catalyst increase the exposure of its abundant acidic (0.078 mmol/g) and basic (0.081 mmol/g) active centers compared with other composites. Therefore, this catalyst exhibits a shorter induction period and better catalytic performance. Furthermore, compared with the impregnation method and mechanochemical method, the hydrothermal coating method produces a greater variety of acid–base active centers in the composite catalyst due to the hydrothermal modifying effect.

## 1. Introduction

As an important organic intermediate in the fields of pesticides, medicine, and animal feed, the most important application of methanethiol (CH_3_SH) is the production of methionine, an essential amino acid for humans and animals [1,2,3,4,5,6]. Based on the used raw materials and synthesis routes, the production process of CH_3_SH can be divided into the methanol–hydrogen sulfide method, methyl chloride–alkali sulfide method, alkali sulfide–dimethyl sulfate method, thiourea–dimethyl sulfate method, methanol–carbon disulfide method, high-sulfur syngas method, and other methods [7,8,9,10,11,12,13,14,15,16,17,18,19,20,21,22,23,24,25]. Among these, the methanol–hydrogen sulfide method has attracted considerable attention due to its higher conversion efficiency and product selectivity [26,27,28].

For the methanol–hydrogen sulfide synthesis process of CH_3_SH, catalyst research has mainly focused on the preparation and modification of molecular sieves and/or metal oxides. The effects of different modifying elements (alkali metals, alkaline earth metals, Cd, Zn, Ni, Co, Nb.) on the structural properties and catalytic performance of molecular sieves (NaY, NaX, MCM-41, ZSM-5, SBA-15, SAPO-18.) have been investigated [29,30,31,32,33,34]. In addition, the effects of different modifying elements (alkali metals, alkaline earth metals, Mo, Nb, Si, Zn, W, La.) on the structural properties and catalytic performance of metal oxides (ZrO_2_, MgO, Al_2_O_3_, TiO_2_, CeO_2_, Nb_2_O_5_, SiO_2_.) have been studied [26,27,35]. The methanol-hydrogen sulfide synthesis route follows a bimolecular acid–base catalysis mechanism [36,37,38]. The acidic centers of the catalyst are favorable for methanol conversion and the generation of dimethyl sulfide (CH_3_SCH_3_) as a by-product, while the basic centers catalyze the dissociative adsorption of hydrogen sulfide and the selective production of CH_3_SH. Methanol is first adsorbed on the acid centers and dissociated to form active methoxy, and then methoxy can react with hydrogen sulfide molecule or dissociated sulfhydryl groups (-SH) to form CH_3_SH (Appendix A) [28]. The formation of -SH contributes to the enhancement of CH_3_SH selectivity. Therefore, effective regulation of the acid–base property of the catalyst contributes significantly to high product selectivity.

To control the catalytic acid–base properties, our group has, in previous studies, proposed the design of composites based on oxides and molecular sieves and prepared a ZrO_2_/NaZSM-5 composite catalyst [37,38]. The construction of various acid–base active centers and a hierarchical mesopore–micropore structure has significantly enhanced the catalytic performance of the material. While the NaZSM-5 molecular sieve has been widely applied because of its various and adjustable active centers and characteristic microporous structure, adjusting the Si/Al ratio of the NaZSM-5 framework and constructing hierarchical porous structures has been pursued to create suitable active centers, strengthen molecular diffusion, and enhance the reactivity and selectivity while simultaneously inhibiting catalyst deactivation. Among the previous modification strategies, the common route involves impregnation and desiliconization/dealumination in acidic or basic environments [39,40,41,42,43,44]. ZrO_2_ is a transition metal oxide with both acidic and basic properties [45,46,47,48,49], in which its monoclinic (m) and tetragonal (t) crystalline phases can be used as catalysts or carriers. Moreover, studies indicate that the surface/interface properties of ZrO_2_ are closely related to the crystal structure [45,50,51]. The effect of compositing ZrO_2_ and NaZSM-5 on the catalytic reaction needs to be further clarified. Therefore, the present paper focuses on the design and preparation of ZrO_2_/NaZSM-5 composite catalysts, and, with emphasis on the ZrO_2_ component, discusses in-depth the effects of the crystal phase structure of ZrO_2_ and the compositing method on the physicochemical properties and catalytic performance of the obtained composite catalysts. The study reveals the structure–performance relationship of the ZrO_2_/NaZSM-5 composite catalysts, which contributes to the fundamental understanding of the structural design of catalysts with a composite phase structure.

## 2. Materials and Methods

### 2.1. Preparation of Catalysts

#### 2.1.1. Preparation of NaZSM-5 Molecular Sieve

NaZSM-5 molecular sieve was synthesized via a hydrothermal method. Typically, 0.03 g NaAlO_2_ (AR, Chengdu Gracia Company, Chengdu, China), 6.5 mL tetrapropylammonium hydroxide (TPAOH, AR, Shanghai Aladdin Company, Shanghai, China) and 15 mL of tetraethoxysilane (TEOS, AR, Shanghai Aladdin Company, Shanghai, China) were dissolved in deionized water to obtain 144 mL of a mixed solution according to the molar ratio of the raw materials TEOS:NaAlO_2_:TPAOH:H_2_O of 180:1:19:8077. After continuous stirring for 30 min, the solution was transferred into an autoclave and then reacted at 180 °C for 48 h under stirring at 120 r/min. Then, the mixture was cooled and centrifuged. The obtained solid was washed three times with deionized water and absolute ethanol. The product was dried at 105 °C for 10 h and calcined in air at 550 °C for 3 h. Finally, 2.5 g NaZSM-5 molecular sieve was obtained.

#### 2.1.2. Preparation of ZrO_2_

Monoclinic zirconia (m-ZrO_2_) was prepared by a liquid-phase precipitation method. Under stirring at room temperature, 13.28 g sodium dodecyl sulfonate (SDS, Tianjin Comeo, AR, Tianjin, China) and 20 g Zr(NO_3_)_4_·5H_2_O (Shanghai McLean Biochemical Technology Co., Ltd., AR, Shanghai, China) were dissolved in 25 mL ethanol–water solution (1:1, *v*/*v*). To this solution, a 0.5 M solution of NaHCO_3_ (Shanghai Aladdin Company, AR, Shanghai, China) was slowly added at 80 °C until a pH of 9 was reached. Stirring continued for 6 h, and the mixture aged for a further 12 h. After cooling to room temperature, the suspension was centrifuged. The solid product was washed three times with deionized water and absolute ethanol, dried at 105 °C for 10 h, and calcined in air at 550 °C for 3 h to obtain m-ZrO_2_.

For the preparation of tetragonal zirconia (t-ZrO_2_), 4.68 g SDS and 13.93 g Zr(NO_3_)_4_·5H_2_O were dissolved in 480 mL ethanol–water solution (1:1, *v*/*v*). Otherwise, the synthesis followed the process described for the preparation of m-ZrO_2_.

#### 2.1.3. Preparation of Composite Catalysts

An m-ZrO_2_/NaZSM-5-HC composite catalyst was prepared via a hydrothermal coating method. A certain amount of 200-mesh m-ZrO_2_ powder was added to the above-mentioned NaZSM-5 precursor solution; the mixture was stirred and transferred into an autoclave. After reaction at 180 °C under stirring at 120 r/min for 48 h, the system was cooled, centrifuged, and separated. After washing three times with deionized water and absolute ethanol, the product was dried at 105 °C for 10 h and calcined in air at 550 °C for 3 h to obtain m-ZrO_2_/NaZSM-5-HC composite catalyst after cooling to room temperature. Unless otherwise specified, the content of m-ZrO_2_ in the composite catalyst was 50 wt.%.

Similarly, a t-ZrO_2_/NaZSM-5-HC composite catalyst was prepared by the same hydrothermal coating method. According to a t-ZrO_2_ content of 50 wt.%, 2.5 g 200-mesh t-ZrO_2_ powder was added to 144 mL NaZSM-5 precursor solution obtained as described in 2.1.1. The subsequent synthesis steps were the same as for the preparation of m-ZrO_2_/NaZSM-5-HC.

An m-ZrO_2_/NaZSM-5-PB composite catalyst with an m-ZrO_2_ content of 50 wt.% was prepared by physical blending of 2.5 g 200-mesh NaZSM-5 and 2.5 g of 200-mesh m-ZrO_2_. 

An m-ZrO_2_/NaZSM-5-IM composite catalyst was prepared using the equal-volume impregnation method. Typically, according to an m-ZrO_2_ content of 50 wt.%, the corresponding Zr(NO_3_)_4_·5H_2_O amount was dissolved in a volume of water equal to that adsorbed by a certain amount of 200-mesh NaZSM-5 at room temperature. After adding the Zr solution, NaZSM-5 was impregnated for 12 h. The product was dried at 105 °C for 10 h and calcined at 550 °C for 3 h. 

The composite m-ZrO_2_/NaZSM-5-MC with an m-ZrO_2_ content of 50 wt.% was prepared by a mechanochemical method. Typically, 2.5 g of 200-mesh NaZSM-5 and 2.5 g of 200-mesh m-ZrO_2_ were poured into the ball-milling jar, and then put into the planetary ball mill (FP4, Shandong Xinhai Group, Shandong, China for grinding at 300 rpm for 10 min (zirconia ball diameter 20 mm). 

### 2.2. Test of Catalysts

The catalytic performance for methanol thiolation was evaluated in a high-pressure fixed-bed reactor using a catalyst amount of 1 g. After a 3 h pre-sulfiding process with hydrogen sulfide, the catalytic reaction was conducted under the following conditions: a hydrogen sulfide/methanol molar ratio of 2:1, a space velocity of 1.5 h^−^^1^, an N_2_ flow rate of 90 mL/min, a reaction pressure of 0.7 MPa, and a reaction temperature of 370 °C. The contents of reactants and products during the reaction were detected online and analyzed with a GC9790II gas chromatograph (Zhejiang Fuli Analytical Instrument Co., Ltd., Zhejiang, China) using a Porapak.Q 1.5 m × 3 mm column with an FID detector and a KB-PLOTQ 30 m × 0.53 mm × 40.00 μm column with an FPD detector. From the obtained data, methanol conversion rate and selectivity and yield of CH_3_SH were calculated.

### 2.3. Characterization of Catalysts

Phase compositions of the obtained catalysts were characterized by a D8-type X-ray diffractometer (XRD, Bruker, Karlsruhe, Germany) using a Cu target at a voltage of 40 kV, a current of 40 mA, a scanning rate of 5°·min^−1^, and a 2θ scanning range of 5–90°. The chemical environment of the catalyst surface elements was analyzed with a K-Alpha Plus X-ray photoelectron spectrometer (XPS, Thermo Fisher, Waltham, MA, USA) using Al-Kα radiation (1486.6 eV), a step size of 0.05 eV, and a power of 150 W. A Zetium multifunctional X-ray fluorescence spectrometer (XRF, Malvern Panalytical Company, Malvern, UK) was used with a power of 4 kW to qualitatively and quantitatively detect the bulk-phase chemical element composition of the catalysts. ^29^Si NMR spectra of samples were characterized and analyzed using an Advance III HD instrument (Bruker, Karlsruhe, German) at a frequency of 119.2 MHz. The catalyst framework structure was characterized by an IS50 infrared spectrometer (FT-IR, Thermo Fisher Scientific, Waltham, MA, USA) using KBr (99.99%) for background calibration. KBr disks were prepared with a mass ratio of sample to KBr of 1:100, and the spectral range was 4000–400 cm^−^^1^. Specific surface areas and pore structure parameters of the catalysts were analyzed and characterized using an ASAP 2020 (M) physical adsorption instrument (Micromeritics Company, Atlanta, GA, USA). After degassing at 250 °C for 3 h, N_2_ adsorption was performed at 77 K. The BET equation was used to calculate the specific surface area of the sample, and the t-plot method was employed to determine the specific surface areas of the micropores (S_mic_) and mesopores (S_mes_). The t-plot method and BJH equation were used to calculate the micropore volume (V_mic_) and mesopore volume (V_mes_), and the total pore volume was calculated when P/P_0_ = 0.995. Acid–base properties of the catalyst surfaces were detected and analyzed by an AutoChem II 2920 automatic chemical adsorption analyzer (Micromeritics, Atlanta, GA, USA) under the following conditions: sample amount of 80 mg; under N_2_ flow (30 mL/min), the sample was activated at 300 °C for 1 h and then cooled to 45 °C for continuous adsorption of NH_3_-He (10 vol% NH_3_) or CO_2_-He (10 vol% CO_2_) mixed gas for 0.5 h; then, the temperature was increased to 700 °C (10 °C/min) for desorption.

## 3. Results

### 3.1. Effects of ZrO_2_ Crystal Phase Structure

Figure 1 shows the catalytic performances of m-ZrO_2_/NaZSM-5-HC and t-ZrO_2_/NaZSM-5-HC composite catalysts prepared by the hydrothermal coating method. Methanol conversion and CH_3_SH selectivity of m-ZrO_2_/NaZSM-5-HC were significantly higher than those of t-ZrO_2_/NaZSM-5-HC. In addition, the reaction induction time of m-ZrO_2_/NaZSM-5-HC is significantly shorter, indicating that this catalyst possesses higher initial reactivity and catalytic performance. The variation in the catalytic performance is related to different physicochemical properties of these composite catalysts caused by the different crystal phase structures of the component ZrO_2_. 

In the XRD patterns of different samples (Figure 2), the diffraction peaks can be assigned to NaZSM-5, m-ZrO_2_, and t-ZrO_2_ according to PDF No.47-0638, PDF No.83-0944, and PDF No.50-1089, indicating that the samples consist of the respective pure phases. While the characteristic diffraction peaks of the two-phase structure, NaZSM-5 and t-ZrO_2_, can be identified in the XRD pattern of t-ZrO_2_/NaZSM-5-HC, careful comparison reveals that the diffraction peaks of NaZSM-5 tend to shift to larger angles. This shift indicates that the inter-planar spacing of NaZSM-5 decreases after the compositing process, which suggests that Zr ions in the t-ZrO_2_ phase partly dissolve and enter the framework of the molecular sieve. It should be pointed out that in the diffraction pattern of m-ZrO_2_/NaZSM-5-HC, the diffraction peaks of the NaZSM-5 crystal phase decrease in intensity and show a more pronounced shift to larger angles. This may be related to the dissolution of more Zr ions from m-ZrO_2_ phase into the framework of NaZSM-5 molecular sieve during hydrothermal synthesis, indicating that the compositing interaction between the two phases has been further strengthened (Appendix A). This analysis result can also be confirmed by the XPS spectra of different composite catalysts (Appendix A). Compared with pure NaZSM-5, the binding energy of Si element in the composite catalysts decreases, and m-ZrO_2_/NaZSM-5-HC demonstrates a more significant reduction in Si binding energy, indicating a strengthened interaction between the two phases in this composite catalyst. 

To further clarify the mechanism influencing the framework of NaZSM-5, the elemental composition of the bulk phase was determined by XRF before and after the compositing process, and the results are listed in Table 1. Interestingly, compared with pure NaZSM-5, the bulk-phase Si/Al ratios of composite catalysts decrease to different degrees; m-ZrO_2_/NaZSM-5-HC exhibits a more significant decrease in Si/Al ratio than t-ZrO_2_/NaZSM-5-HC and also shows the highest bulk-phase Zr/Si ratio. It is assumed that more Zr ions in this sample have been incorporated into the framework of the molecular sieve and caused the decrease in Si/Al ratio.

To further investigate the changes in the chemical environment of Si in the zeolite framework, solid-state ^29^Si NMR measurements were conducted on m-ZrO_2_/NaZSM-5-HC and m-ZrO_2_/NaZSM-5-PB. The results are shown in Figure 3. The Si/Al ratio on the framework of m-ZrO_2_/NaZSM-5-HC is only 113, which is lower than that of m-ZrO_2_/NaZSM-5-PB (164). This trend corroborates the above-mentioned assumption. Zr ions dissolved from m-ZrO_2_ in a specific hydrothermal environment enter the framework of the molecular sieve, which is the direct reason for the lower Si/Al ratio.

Specific surface areas and pore structure parameters of different catalysts are listed in Table 2 and Appendix A. It can be seen that pure NaZSM-5 and ZrO_2_ mainly exhibit structural characteristics of micropores and mesopores, respectively, while pure m-ZrO_2_ show a micro-mesoporous composite structure. The NaZSM-5 molecular sieve has the highest specific surface area (365 m^2^/g); the introduction of mesoporous ZrO_2_ results in a meso-microporous structure of two composite catalysts, but the m-ZrO_2_/NaZSM-5-HC composite catalyst has a larger specific surface area (360 m^2^/g) and pore volume (0.52 cm^3^/g) than t-ZrO_2_/NaZSM-5-HC. Its specific surface area approaches, and its pore volume even exceeds, that of pure NaZSM-5 molecular sieve, which provides the possibility for the composite catalyst to expose more active sites.

NH_3_/CO_2_-TPD spectra of different catalysts are shown in Figure 4. Peaks in the temperature range of 50–600 °C, are expressed by the concentrations of weak (<200 °C), medium-strong (200–400 °C) and strong (400–600 °C) acid or base sites. In Figure 4a, the pure NaZSM-5 molecular sieve has the largest total acid content but only weak acid centers. The total acid content of the composite catalysts ranges between that of ZrO_2_ and that of NaZSM-5, and both weak and medium-strong acid centers are present. Interestingly, the composite catalysts have both weak and medium-strong base centers, and their total base content far exceeds those of pure ZrO_2_ and NaZSM-5 (Figure 4b). The large increase in total base content of the composite catalysts is related to the reduction in the framework Si/Al ratio caused by the desiliconization of NaZSM-5 [52,53]. As can be seen from Appendix A, no strong acid/base centers are observed in all catalysts. Moreover, the amount of medium strong acid/base of the composite catalysts are basically the same as that of weak acid/base.

It should be pointed out that the total acid content and base content of m-ZrO_2_/NaZSM-5-HC are both higher than those of t-ZrO_2_/NaZSM-5-HC, which is related to a greater change in Si/Al ratio during the preparation of these composite catalysts. The distribution of generous base centers in m-ZrO_2_/NaZSM-5-HC contributes significantly to the highly selective production of CH_3_SH.

In addition, the O 1s XPS spectra of the composite catalysts (Figure 5) indicate different degrees of oxygen vacancy defects in these catalysts. Combined with the ^29^Si NMR analysis result, it is speculated that the formation of oxygen vacancies is caused by the combination of active protons of the silanol groups (Si-OH) generated by the removal of Si from the framework and the hydroxyl groups present on the ZrO_2_ component [42,54,55,56,57]. However, m-ZrO_2_/NaZSM-5-HC contains more oxygen vacancies as defects, which is closely related to the change in Si/Al ratio of the framework of the molecular sieve in the composite catalyst. Considering the acid–base catalysis mechanism of the methanol-hydrogen sulfide reaction to produce CH_3_SH, in addition to the Brønsted (B) acid sites of the molecular sieve, oxygen vacancies as a special Lewis (L) acid are also important for the adsorption and dissociation of methanol molecules. The synergistic B-L acid active centers constructed by the compositing effect facilitate the dissociation and transformation of methanol molecules under milder conditions [36,37,38].

In conclusion, the higher initial reactivity and catalytic performance of m-ZrO_2_/NaZSM-5-HC can be attributed to the exposure of more active sites due to its large specific surface area, the construction of a characteristic meso-microporous structure, and its different acid-base centers.

### 3.2. Effect of the m-ZrO_2_ Compositing Method

To further investigate the effects of the m-ZrO_2_ compositing method on the compositing effect, we compared and analyzed the effect of the hydrothermal coating method (m-ZrO_2_/NaZSM-5-HC), the impregnation method (m-ZrO_2_/NaZSM-5-IM), the mechanochemical method (m-ZrO_2_/NaZSM-5-MC), and the physical blending method (m-ZrO_2_/NaZSM-5-PB) on the physicochemical properties and catalytic performance of these composite catalysts.

XRD patterns of different composite catalysts are shown in Figure 6. The characteristic diffraction peaks of the two-phase structure, NaZSM-5 and m-ZrO_2_, can be observed in all composite catalysts. However, compared with those of m-ZrO_2_/NaZSM-5-PB, the diffraction peaks of NaZSM-5 in the other composite catalyst samples exhibited lower intensity and a shift to larger angles to different extents, indicating that all three preparation methods promote the dissolution of a certain amount of Zr ions from the m-ZrO_2_ phase and their incorporation into the framework of the molecular sieve. Peak intensity and degree of peak shifting decrease in the order m-ZrO_2_/NaZSM-5-HC > m-ZrO_2_/NaZSM-5-IM > m-ZrO_2_/NaZSM-5-MC, which is consistent with the trend of the change in Si/Al ratio of the molecular sieve for different composite samples (Table 3). The results indicate that the m-ZrO_2_/NaZSM-5-HC sample prepared by the hydrothermal coating method exhibits a more prominent compositing effect, followed by the impregnation method and the mechanochemical method. The physical blending method only produces a macroscopic mixture of two phases, which exist independently and exhibit the weakest interaction between them among the compared composites.

This trend has been further verified by the trend in binding energies in the XPS data of Si and Zr elements in composite samples (Appendix A). Compared with m-ZrO_2_/NaZSM-5-PB, the binding energies of Si in the other composite samples decrease and those of Zr increase by varying degrees; the trends are consistent with those of XRD and XRF data, indicating that the composite catalyst sample obtained by the hydrothermal coating method exhibit significantly enhanced interactions. 

Interestingly, m-ZrO_2_/NaZSM-5-PB has the smallest total acid and total base content, while the trend in the change of acid–base content of other composite samples is consistent with their trends of Si binding energy, that is, m-ZrO_2_/NaZSM-5-HC has the highest acid–base content (Figure 7). The result is related to the different degrees of desiliconization caused by different compositing methods. The hydrothermal coating modification process changes the Si/Al ratio of the NaZSM-5 framework to the greatest extent, resulting in the modification of the framework structure. The characteristic FT-IR peaks (Appendix A) of the framework of the molecular sieve at 545 cm^−1^ and 778 cm^−1^ gradually decrease in intensity, while the absorption peak at 1212 cm^−1^ becomes obvious gradually. It is noted that m-ZrO_2_/NaZSM-5-PB has only weak acid and weak base centers, which is attributed to the fact that pure NaZSM-5 and pure m-ZrO_2_ also only possess weak acid and weak base centers (Figure 4). The m-ZrO_2_/NaZSM-5-MC and m-ZrO_2_/NaZSM-5-IM have similar active sites as m-ZrO_2_/NaZSM-5-PB (Appendix A). As mentioned above, although the composite approach of impregnation and mechanochemical methods can also permit small amounts of Zr ions in m-ZrO_2_ to enter the molecular sieve skeleton, this slight structural change does not seem to be enough to change the type of active centers. However, since m-ZrO_2_/NaZSM-5-HC features weak acid, medium-strong acid, weak base, and medium-strong base centers, the hydrothermal coating process can apparently modify the type of acid-base centers in the composite catalyst. 

In addition, the specific surface area of m-ZrO_2_/NaZSM-5-IM is slightly higher than that of m-ZrO_2_/NaZSM-5-PB (Table 3 and Appendix A). As a possible explanation, the impregnation process might lead to the desiliconization of the molecular sieve and then form a new pore system. Interestingly, compared with m-ZrO_2_/NaZSM-5-PB, both specific surface area and pore volume of m-ZrO_2_/NaZSM-5-MC decrease. Based on further analysis, this decrease was mainly caused by the significant reduction in specific surface area and pore volume of mesopores, which indicates that, although the mechanochemical method changes the Si/Al ratio of NaZSM-5 molecular sieve, the mesoporous structure has been partly destroyed by the ball–milling process. In contrast, m-ZrO_2_/NaZSM-5-HC prepared via the hydrothermal coating method has a lower specific surface area of micropores due to the desiliconization of the microporous framework of the molecular sieve, while the specific surface area of mesopores increases considerably, resulting in a significant increase in both total specific surface area and pore volume.

It can be observed from Figure 8 that the change trend of catalytic performance of composite samples is consistent with that of acid-base properties. Further, m-ZrO_2_/NaZSM-5-HC prepared by hydrothermal coating has medium strong acid-base center, which is the key to greatly improve the catalytic performance. In addition, the structural characteristics of the catalysts also have an important impact on the catalytic performance. The composite m-ZrO_2_/NaZSM-5-PB shows the worst catalytic performance since it is only a physical mixture of NaZSM-5 and m-ZrO_2_, without a special modification of the molecular sieve. For m-ZrO_2_/NaZSM-5-MC, the mechanochemical process modifies the Si/Al ratio of the framework of the molecular sieve to a certain extent, but the accompanying damage to the mesoporous system results in a decrease in both specific surface area and pore volume of the composite catalyst, which renders the enhancement of its catalytic performance insignificant. For m-ZrO_2_/NaZSM-5-IM, the impregnation process increases the modification of the framework structure of the molecular sieve, resulting in slightly enhanced catalytic performance. Nevertheless, the catalytic performance of all composite catalysts is higher than that of NaZSM-5.

In conclusion, by applying mesoporous ZrO_2_ in the modification of the molecular sieve, a part of the Zr^4+^ ions dissolves and enters NaZSM-5 molecular sieve under hydrothermal conditions to cause desiliconization of its framework, thereby enlarging the specific surface area of the composite catalyst and increasing the total acid–base content. Additionally, the hydrothermal modification effect also produces different types of active centers; therefore, m-ZrO_2_/NaZSM-5-HC exhibits the best catalytic performance. Under the reaction conditions of a hydrogen sulfide/methanol molar ratio of 2:1, a space velocity of 1.5 h^−1^, an N_2_ flow rate of 90 mL/min, a reaction pressure of 0.7 MPa, a reaction temperature of 370 °C, a methanol conversion rate of 92%, a CH_3_SH selectivity of 95%, and a CH_3_SH yield of 87.4% can be achieved. 

## 4. Conclusions

Based on the design idea of using a metal oxide in the modification of a molecular sieve, this paper focuses on the preparation of ZrO_2_/NaZSM-5 composite catalysts. In combination with the methanol–hydrogen sulfide reaction to produce CH_3_SH following a bimolecular acid–base catalysis mechanism, this paper investigates the effects of crystal phase structure and compositing method of ZrO_2_ on the physicochemical properties and catalytic performance of composite catalysts. By comparing the structure–performance relationships of different composite catalysts, it is concluded that in the composite prepared by a hydrothermal coating method, Zr^4+^ ions are more like to dissolve from m-ZrO_2_ than from t-ZrO_2_, which then enter the molecular sieve, resulting in a higher degree of desiliconization of the NaZSM-5 framework. Consequently, the obtained m-ZrO_2_/NaZSM-5 composite catalyst demonstrates the best catalytic performance due to its largest specific surface area, pore volume, and acid–base contents. In addition, compared with the compositing effect of the impregnation and mechanochemical methods, the hydrothermal coating method also produces several types of acid–base active centers in the composite catalyst due to the hydrothermal modification. Regarding catalytic performance, these above-mentioned modifications are favorable for methanol thiolation. In view of the adjustable characteristics of NaZSM-5 molecular sieves, this study focuses on the modification strategy based on the hydrothermal coating process to enhance catalytic efficiency and reveals the mechanism of compositing and modifying molecular sieves with ZrO_2_, which enriches the modification methods for molecular sieves.

## Figures and Tables

**Figure 1 nanomaterials-12-01803-f001:**
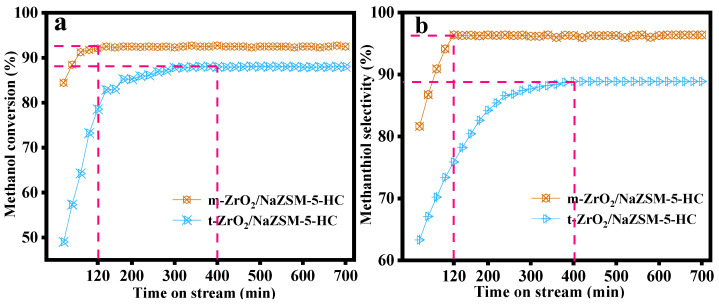
Catalytic performance of m-ZrO_2_/NaZSM-5-HC and t-ZrO_2_/NaZSM-5-HC composite catalysts: (**a**) methanol conversion rate and (**b**) selectivity to CH_3_SH.

**Figure 2 nanomaterials-12-01803-f002:**
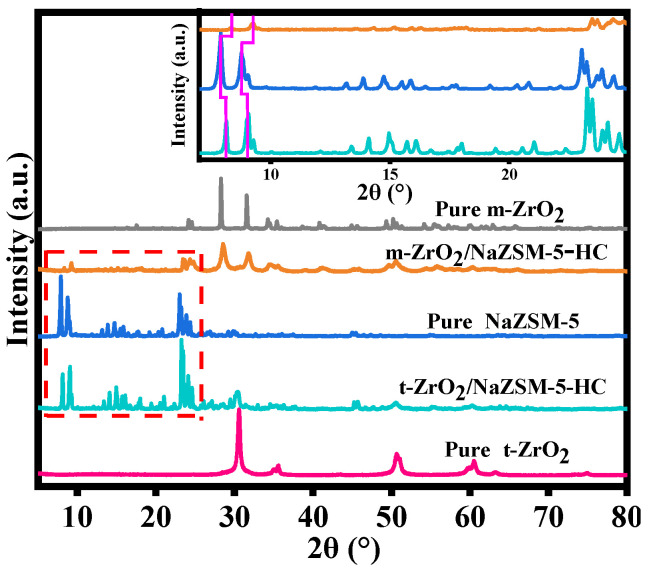
XRD patterns of different catalysts.

**Figure 3 nanomaterials-12-01803-f003:**
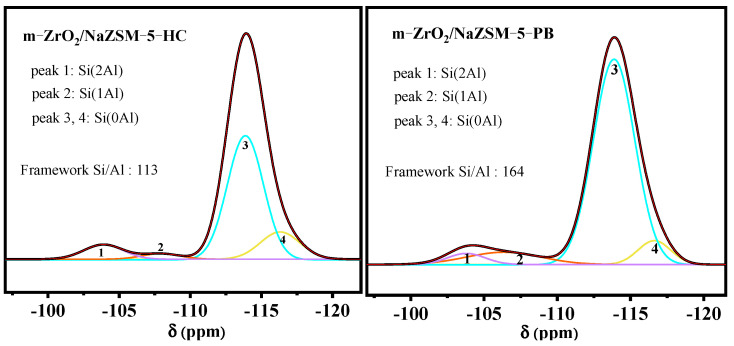
^29^Si NMR spectra of m-ZrO_2_/NaZSM-5-HC and m-ZrO_2_/NaZSM-5-PB composite catalysts.

**Figure 4 nanomaterials-12-01803-f004:**
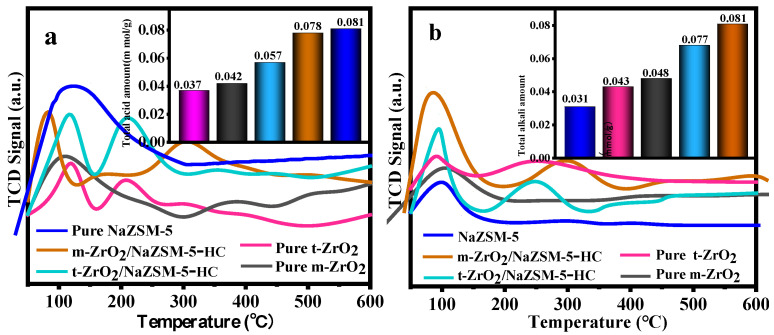
TPD spectra of different catalysts: (**a**) NH_3_-TPD and (**b**) CO_2_-TPD.

**Figure 5 nanomaterials-12-01803-f005:**
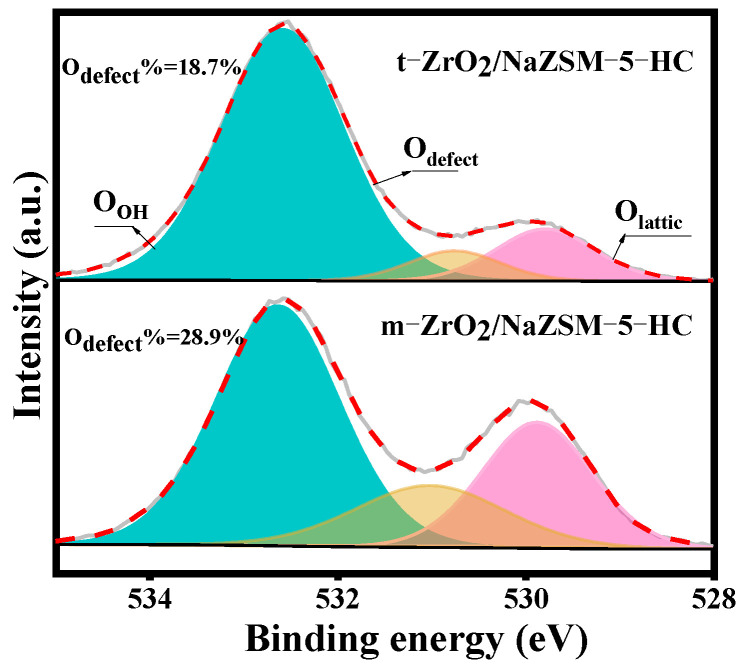
O 1s XPS spectra of t-ZrO_2_/NaZSM-5-HC and m-ZrO_2_/NaZSM-5-HC composite catalysts.

**Figure 6 nanomaterials-12-01803-f006:**
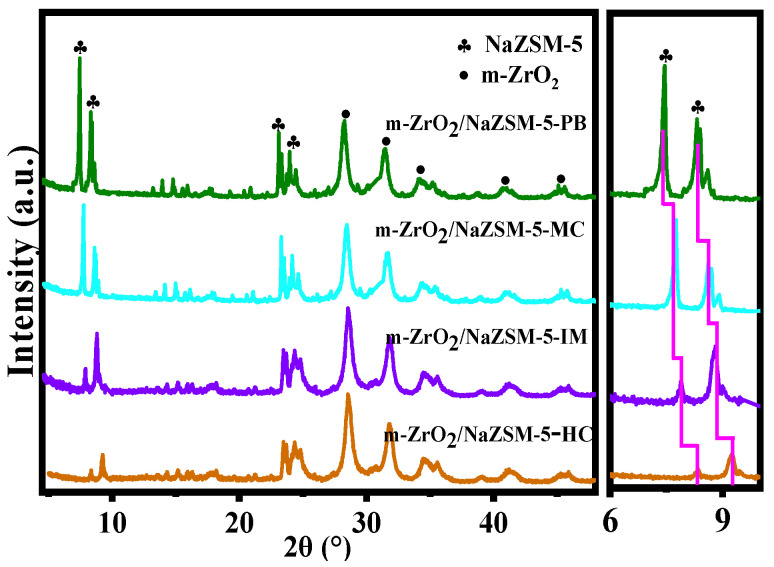
XRD patterns of different composite catalysts.

**Figure 7 nanomaterials-12-01803-f007:**
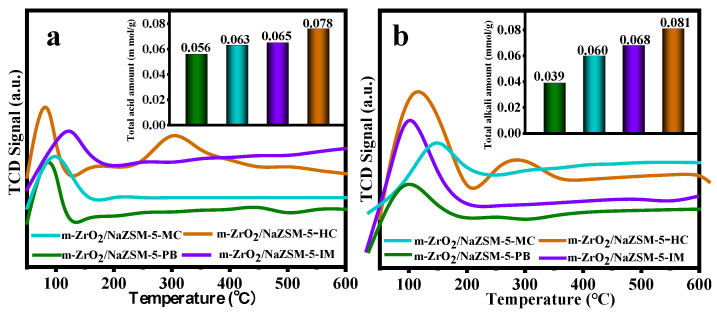
TPD spectra of different composite catalysts: (**a**) NH_3_-TPD and (**b**) CO_2_-TPD.

**Figure 8 nanomaterials-12-01803-f008:**
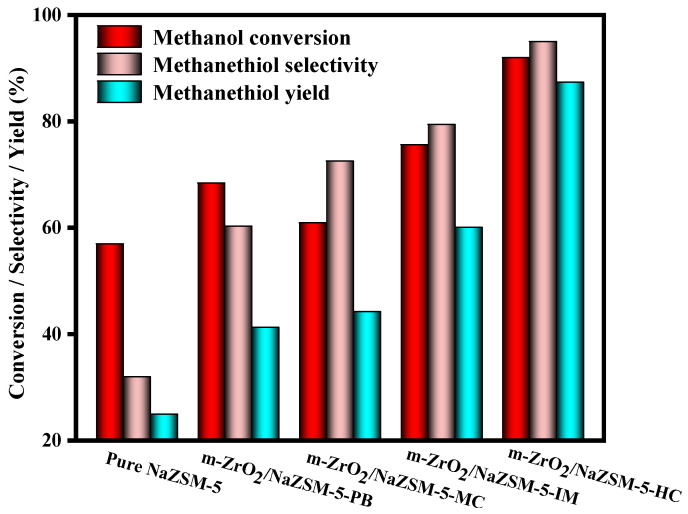
Catalytic performance of different composite catalysts.

**Table 1 nanomaterials-12-01803-t001:** Elemental composition of different catalysts.

Samples	Element Content (wt.%)					Bulk Zr/Si Bulk Si/Al	
	Si	Al	Zr	O	Na		
m-ZrO_2_/NaZSM-5-HC	17.09	0.106	36.53	43.22	0.106	2.14	161
t-ZrO_2_/NaZSM-5-HC	20.42	0.117	31.83	44.58	0.115	1.56	174
NaZSM-5	35.89	0.20	-	60.75	0.19	-	179

**Table 2 nanomaterials-12-01803-t002:** Specific surface areas and pore structure parameters of different catalysts.

Composites	BET Surface Area (m^2^/g)			Pore Volume (cm^3^/g)		
	S_BET_	S_mic_	S_mes_	V_t_	V_mic_	V_mes_
Pure NaZSM-5	365	330	35	0.41	0.34	0.07
m-ZrO_2_/NaZSM-5-HC	360	126	234	0.52	0.17	0.35
t-ZrO_2_/NaZSM-5-HC	239	114	125	0.43	0.12	0.31
Pure m-ZrO_2_	134	40	94	0.47	0.21	0.26
Pure t-ZrO_2_	184	-	184	0.50	-	0.50

**Table 3 nanomaterials-12-01803-t003:** Pore structure characteristics and elemental compositions of different composite catalysts.

Catalyst	BET Surface Area (m^2^/g) ^a^			Volume (cm^3^/g) ^a^			XRF ^b^ Si/Al
	S_BET_	S_mic_	S_mes_	V_t_	V_mic_	V_mes_	
m-ZrO_2_/NaZSM-5-PB	268	154	114	0.42	0.28	0.14	179
m-ZrO_2_/NaZSM-5-MC	235	152	83	0.40	0.31	0.09	170
m-ZrO_2_/NaZSM-5-IM	276	161	115	0.42	0. 30	0.12	167
m-ZrO_2_/NaZSM-5-HC	360	126	234	0.52	0.17	0.35	161

^a^ Measured by N_2_ sorption at 77K. ^b^ Determined by XRF.

## Data Availability

The data that support the findings of this study are available upon reasonable request.

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
