# Peer review of "Role of Zirconia in Oxide-Zeolite Composite for Thiolation of Methanol with Hydrogen Sulfide to Methanethiol"

_nanomaterials, 2022, doi:10.3390/nano12111803_

Round 1
Reviewer 1 Report
This manuscript deals with the preparation, characterization, and catalytic performance in the methanethiol synthesis of ZrO2-modified NaZSM-5 zeolites. I recommend the publication after minor revision, according to the following comments/suggestions.
- p. 2, lines 50-51: it's not clear to me how methoxy groups can be formed through methanol-acid sites interaction. A reaction scheme should be proposed for sake of clarity.
- Table 1: check the columns' headers.
- p. 6, line 216: regarding "Si framework", maybe one should more appropriately speak of changes in the chemical environment of Si in the zeolite framework.
- p. 6, lines 218-220: please, make the following sentence clearer: Compared with m‑ZrO2/NaZSM‑5-PB with a Si/Al ratio of 164, the Si/Al ratio of the framework of m‑ZrO2/NaZSM‑5-HC is with 113 much lower.
- p. 6, lines 227-233: this part is redundant, since it reports what already said (see pag. 5, lines 188-198).
- p. 6, lines 235-236: "According to the data, pure NaZSM-5 and ZrO2 mainly exhibit structural characteristics of micropores and mesopores, respectively " is not true for m-ZrO2, for which a remarkable contribution of microporosity to the specific surface area (30 %) and pore volume (45 %) can be observed.
- p. 7, lines 247-248: please, define for both acidity and basicity the range of desorption temperatures used to rank the sites as weak, moderate and strong. Moreover, it should be interesting to estimate the amount of acid and base sites of different strength for all the samples.
- p. 7, lines 250-252: in the case of m-ZrO2/NaZSM-5-HC, what is the reason why its acidity (0.078) is very similar to that of NaZSM-5 (0.081)?
- p. 7, lines 254-261: as in the case of XRD, also these results on the textural properties have already been discussed (see p. 6, lines 234-242).
- pag. 9, lines 332-335: what does it mean? that the MC and IM composites exhibit acid and basic sites of the same strength than the PB sample? As already said, it is important to clearly indicate the temperature ranges for distinguishing the sites of different strength, since shifts of the peaks at higher temperatures is sometimes observed. Furthermore, it was previously said that "However, compared with those of m-ZrO2/NaZSM-5-PB, the diffraction peaks of NaZSM-5 in the other composite catalyst samples exhibited lower intensity and a shift to larger angles to different extents, indicating that all three preparation methods promote the dissolution of a certain amount of Zr ions from the m-ZrO2 phase and their incorporation into the framework of the molecular sieve." (p. 8, lines 299-302); thus, the reason why no remarkable changes in the acid-base properties have occurred (at variance with the HC composite) should be explained.
- p. 10, lines 357-358: In the discussion, it seems that the different catalytic performance depends only on the different textural features of the catalysts prepared by the different synthesis procedure. However, they differs also for acidity and basicity. This aspect should be discussed in more details, also in relation to the role of sites of different strength.
Author Response
We are very grateful to the reviewer for the useful suggestion,we have revised the manuscript carefully.
Please check the attachment.

Reviewer 2 Report
General Comments
In this work, the NaZSM-5 molecular sieve was modified with zirconium dioxide (ZrO2) by different methods (hydrothermal, impregnation and mechanochemical method) to find the best one. The authors show that hydrothermal coating process produces a greater variety of acid-base active centers in the composite catalyst due to the hydrothermal modifier effect. The effects of active site crystal phase structure (monoclinic and tetragonal zirconia) and composition method on the physicochemical properties is studied also, revealing a structure-performance relationship of these compound catalysts. The textural, acid basic properties and surface characterization of the materials are very interesting and allow study the catalytic behaviour of the materials. This was a very interesting read for me. Overall, the paper is comprehensive and well-written. It makes a nice contribution to the field, and I recommend publication.
Comments:
- More details of mechanochemical method must be incluyed ( instruments, time, speed,…)
- The definition of crystal phase m or t must be on the begging. To enhance the structure of the paper, the explanation of this phase must be done before the catalytic activity.
- The N2 isotherm and PSD must be show in supplementary.
- In the figures 4 and 7, the base line is different for each sample and the calculation of mmol seems difficult so, it is possible to check the integration of them is right?
- The morphological study with images could be very interesting to enhance the quality of the work.
- It is possible to compare also the catalytic activity of the molecular sieve in the same condition?
Author Response
We are very grateful to the reviewer for the useful suggestion, we have revised the manuscript carefully.
Please check the attachment
